# Influence of Gelatin-Chitosan-Glycerol Edible Coating Incorporated with Chlorogenic Acid, Gallic Acid, and Resveratrol on the Preservation of Fresh Beef

**DOI:** 10.3390/foods11233813

**Published:** 2022-11-26

**Authors:** Jinhao Zou, Xueming Liu, Xuping Wang, Huaigu Yang, Jingrong Cheng, Yaosheng Lin, Daobang Tang

**Affiliations:** Key Laboratory of Functional Foods, Ministry of Agriculture and Rural Affairs, Guangdong Key Laboratory of Agricultural Products Processing, Sericultural & Agri-Food Research Institute Guangdong Academy of Agricultural Sciences, Guangzhou 510610, China

**Keywords:** fresh beef, edible coating, polyphenols, meat quality

## Abstract

Chlorogenic acid (CA), gallic acid (GA), and resveratrol (RES) were added to a gelatin (GEL)-chitosan (CHI)-glycerol (GLY) edible coating, and their effects on the coating of fresh beef preservation were investigated. The results revealed that CA had the most significant improvement effect on fresh beef preservation. The combination of GEL-CHI-GLY-CA preserved the color of the beef better and delayed the increase of the total volatile base nitrogen, even though its total phenolic content decreased at a faster rate during beef preservation. GA also improved the preservation effect as on the 12th day of storage, the beef samples treated with GEL-CHI-GLY-GA had the lowest thiobarbituric acid reactive substances (0.76 mg Malondialdehyde (MDA)/kg) and total viable count (6.0 log cfu/g). On the whole, though RES showed an improvement on beef preservation, the improvement was not as good as the other two polyphenols. After 12 days of storage, the beef samples treated with GEL-CHI-GLY-RES had a higher pH value (6.25) than the other two polyphenol treatmed groups. Overall, the three polyphenol-added combinations increased the shelf life of beef by approximately 3–6 days compared to the control group (treated GEL-CHI-GLY with distilled water).

## 1. Introduction

Fresh beef is rich in protein, fat, water, and other nutrients, but is easily contaminated and spoiled by microorganisms during storage, resulting in a decline of meat nutrition, flavor, color, texture, and sensory properties [1,2]. At present, many methods for preserving fresh beef have been proposed, including physical [3], chemical [4], biological [5], and hurdle technology [6]. Edible films or coatings are a promising food preservation barrier technology that has been proposed in recent years. Such films have been studied and applied in aquatic products [7], fruits [8], and vegetables [9]. These films or coatings preserve fresh food through the gelling properties of proteins, polysaccharides and lipids, and the antibacterial properties of some natural active substances [2,7]. Gelatin (GEL), chitosan (CHI), and glycerol (GLY) are common base materials used to produce edible films and coatings [7,10]. It has been demonstrated that GEL-CHI-GLY-based edible coatings can improve beef color stability [11], with monolayer and bilayer coatings having similar effects in maintaining beef color and improving microbial safety [12]. However, edible films and coatings prepared using only these materials had a minimal impact on meat preservation, and even though increasing the thickness of the coating helps to prevent the penetration of oxygen, the anti-oxidative and anti-bacterial properties of the coating remain weak [13,14,15]. Therefore, some natural active substances such as essential oils [6,16], plant extracts [1], and lysozyme [5,17] have been added to edible films and coatings to improve their antioxidant and antibacterial properties.

Polyphenols are a class of phytochemicals with potential health benefits due to their natural, antioxidant, and antibacterial properties [18]. Plant extracts containing polyphenols are useful as a natural preservative for food preservation. Stinging nettle extract [1], green tea aqueous extract [19], *Rumex tingitanus* L extracts [20], and spice extracts (*Syzygium aromaticum* (L.) Merrill et L.M. Perry, *Cinnamomum cassia* (L.) J. Presl, *Origanum vulgare* L., *Brassica nigra* L.) [21] were all reported to inhibit the growth of microorganisms and have antioxidant effects in meat preservation, with the main active components being polyphenolic compounds. Nonetheless, most polyphenols in the extracts are complex mixtures, making the role of individual phenolic compounds in meat preservation unknown [22]. Chlorogenic acid (CA), gallic acid (GA), and resveratrol (RES) are single phenolic substances found in various fruits, vegetables, traditional Chinese medicinal plants, and spices that are known to be natural antioxidants with excellent antibacterial activity [14,23,24]. Among them, CA has been combined with GEL to prepare an edible coating material for preserving the freshness of seafood [14]. Similarly, GA [24] and RES [16] are effective preservation additives used in edible coatings applied to pork. It is hypothesized that these three polyphenols could be used as effective additives to improve the preservation effect of a GEL-CHI-GLY edible coating when applied to beef.

This study aimed to investigate the improvement effect of three polyphenols (CA, GA, and RES) in a GEL-CHI-GLY edible coating on the preservation of fresh beef. The changes in color, physical and chemical indexes, texture, and sensory characteristics of beef were measured during preservation to provide basic data for potential application of polyphenol preservative additives in meat preservation.

## 2. Materials and Methods

### 2.1. Materials

Twenty-five kg of fresh rib eye beef was purchased from a local market (Guangzhou, China) 24 h after slaughter. All microbial mediums and chemical agents (analytical grade, purity of ≥ 99%) were provided by Sinopharm Chemical Reagent Co., Ltd. (Shanghai, China). Gallic acid (GA), chlorogenic acid (CA), resveratrol (RES), chitosan (CHI), gelatin (GEL), and glycerol (GLY) (purity of ≥ 99%) were obtained from Aladdin (Shanghai, China).

### 2.2. Preparation of Edible Coating

The edible coatings were prepared, as previously reported [11]. Briefly, GEL was dissolved in 1000 mL of distilled water, heated in a boiling water bath for 10 min, then added to CHI (pre-dissolved in 0.6% acetic acid) and GLY, stirred with a thermostatic magnetic stirrer (Model MYP11-2, Shanghai Meiyingpu Instrument Manufacturing Co., Ltd., China) at 1000 rpm at 45 °C for 2 h. Subsequently, 0.5% polyphenols (GA, CA, and RES) were then added to the mixture and cooled to room temperature for later use. The final mass fractions of GEL, CHI, GLY, and either one of the polyphenols (GA, CA, RES) in the solutions were at 4, 1, 4, and 0.5%, respectively.

### 2.3. Beef Treatment

The beef was treated using the method outlined by Alirezalu [1]. The rib eye beef was processed on a super clean workbench (SW-CJ-2D, Suzhou Purification Equipment Co., Ltd., Suzhou, China) and divided into 90 pieces (each one ~ 200g and 8 × 5 × 1 cm^3^) using a pre-sterilized knife and cutting board, and 18 pieces of beef were randomly selected as a group. The pieces of beef were soaked separately in the edible coatings prepared in Section 2.2 for 5 min, after which they were removed and hung on a hook for 1 min (the film formed was about 0.2 mm). The control group consisted of beef samples immersed in pre-sterilized distilled water. The processed beef was placed into single-use plastic lunch boxes in a fresh-keeping display case, equipped with a photoperiod controller, and kept at 4 °C to simulate a supermarket retail environment. Three pieces of beef were taken from each group, and their color, pH, total viable count (TVC), total phenolic content (TPC), total volatile base nitrogen (TVB-N), lipid oxidation, and texture of beef were measured at 0 (about 2 h after coating), 2, 4, 6, 9, and 12 days of preservation. Sensory evaluations were performed at the beginning and end of the preservation process.

### 2.4. Determination of Color Values and Acquisition of Beef Images

The color of the samples was measured using a hand-held colorimeter (Model CR-400, 11 mm diameter aperture, Illuminant C, 2° observer, blooming time 3 s, H.J.Unkel Limited, America), and the *L** (lightness), *a** (redness), and *b** (yellowness) values were recorded. Before determining physical and chemical indicators, images of beef were captured using a 12 M-pixel resolution camera (the camera was located about 10 cm above the sample, and the exposure time was 1 s).

### 2.5. Determination of pH Values

The pH value of the samples was determined using the method outlined by Alirezalu [1]. A pH meter (Model FE28, Mettler Toledo, Switzerland) equipped with a pH electrode and temperature probe, pre-calibrated with pH 4.00 and 7.00 buffers, was used to measure the pH of the beef. The beef samples were mixed with potassium chloride solution (0.1 mol/L) at a ratio of 1:10 (*w/v*) prior to pH determination.

### 2.6. Determination of Total Viable Count

The TVC of the samples was determined using the method outlined by Zhang [25]. The plate counting agar (PCA) was used to measure the TVC using the pouring plate method, with the incubation temperature at 36 ± 1 °C, and the incubation time at 48 ± 2 h. The results were counted as log colony forming units (cfu)/g of beef sample.

### 2.7. Determination of Total Phenolic Content

The TPC of the samples were determined using the method outlined by Alirezalu [1]. Briefly, a total of 50 g of beef sample was mixed with 100 mL of distilled water and boiled in a water bath for 20 min. After cooling and filtering of the sample, 0.5 mL of the filtrate along with 0.5 mL of folin-ciocalteau, 5 mL of 7% sodium carbonate solution, and 6 mL of distilled water were placed in a colorimetric tube, vortexed and mixed, and then placed in a dark place to react for 1 h. The absorbance was measured at 760 nm using a UV-VIS spectrophotometer (Model UV-1800, Mettler Toledo, Switzerland). The results were reported as mg of GA equivalent (GAE)/100 g of beef sample.

### 2.8. Determination of Total Volatile Base Nitrogen

The TVB-N value of the samples was determined using the method outlined by Alirezalu [1]. Briefly, in a 150 mL digestion tube, a total of 10 g of beef sample, 75 mL of distilled water, and 1 g of magnesium oxide were weighed. It was then placed in an automatic Kjeldahl nitrogen analyzer for distillation and titration (Model KT8000, Foss, Denmark). The TVB-N value of the sample was expressed as mg/kg beef sample.

### 2.9. Determination of Thiobarbituric Acid Reactive Substances

The thiobarbituric acid reactive substances (TBARS) value of the samples was calculated using the method outlined by Zhang [26]. A 5 g beef sample and 50 mL 7.5% trichloroacetic acid (containing 0.1% disodium EDTA) were mixed in a centrifuge tube, homogenized for 1 min (10,000 rpm) with a homogenizer (Model T25, Ika, Germany), centrifuged at 4000 rpm for 10 min and filtered with double-layer quantitative slow-speed filter paper. The 5 mL of the prepared filtrate was mixed with 5 mL of 0.02 mol/L thiobarbituric acid solution in a glass tube in a boiling water bath for 30 min. After being removed and cooled, the absorbance was measured at 532 nm, with the TBARS value expressed as mg malondialdehyde/kg of beef sample.

### 2.10. Determination of Texture

The samples were cut into small pieces (length × width × thickness, about 2 cm × 2 cm × 1 cm) and subjected to a texture profile analysis (TPA) test using a texture analyzer (Model TA-XT. PLUS, Stable Micro Systems, UK). Each sample was tested five times with a P/50 probe. The test conditions were as follows: the pre-test rate was 4.0 mm/s, the test rate was 2.0 mm/s, the post-test rate was 2.0 mm/s, the degree of compression was 40%, and the trigger force was 5 g. Beef samples were tested for hardness, springiness, and chewiness.

### 2.11. Sensory Evaluation

The sensory evaluation of the raw beef samples raw was carried out following the method described by Zhang [25]. The sensory panel consisted of 10 trained individuals (four male and six females aged between 22 and 45 years) who were all familiar with the main sensory evaluation techniques and had prior experience with meat evaluation. Each panelist took turns entering a separate hut under white light for evaluation. The panelist evaluated beef by observing color, odor, and texture. The color, odor, and texture were scored using a 9-point hedonic scale. Samples with an overall score of 5.4 or higher were considered acceptable. Sensory sessions were held using samples stored for 0 and 12 days. Three samples of beef from each of the 5 treated groups were tested by 10 panelists in 2 sessions, based on three traits (color, odor, and texture). The samples were evaluated on a hedonic scale (score > 7.2: like, 7.2 ≥ score ≥ 5.4: acceptable, score < 5.4: unacceptable). Each panelist was given approximately 10 min to complete the assessment.

### 2.12. Statistical Analysis

Three independent experimental trials (replications) were conducted. A mixed model using IBM SPSS 20.0 was applied to analyze the data, with treatment and storage time as fixed effects, and with sample as a random term including the interaction between these effects. For the sensory traits, panelist was also included as a random term. Significant differences (*p* < 0.05) among the means were identified using Duncan’s multiple range and reported with their ± standard error values (S.E). The figures were processed using Origin 8.0.

## 3. Results and Discussion

### 3.1. Color Evaluation

The specific parameters of beef color were measured as shown in Table 1. The *L**, *a**, and *b** values of beef samples from five different treatment groups were significantly different during the storage period (*p* < 0.05). The *L** values of the beef samples in the T2 treatment were lower on day 0 of storage than that of T1, consistent with the findings of Alirezalu (coating-treated beef samples had lower *L** values on the first day than the control group) [1]. However, *L** values of the beef samples in the three polyphenol-supplemented groups (T3, T4, and T5) were significantly higher than those in T1, indicating that the addition of polyphenols could improve the transparency of GEL-CHI-GLY coatings. Furthermore, the addition of polyphenols significantly protected the lightness of the beef samples during the storage process (*L** values for T3 and T5 groups were 42.1 and 42.7 after 12 days of storage, respectively). Interestingly, the *a** values of T1 samples decreased throughout the storage process, whereas the *a** value of these groups (T2, T3, T4, and T5) showed a slight upward trend in the early storage period, followed by a downward trend. This could be due to the condensation of quinone compounds formed by the oxidation of polyphenols to form dark compounds that improve the redness of beef samples [27], or the antioxidant capacity of polyphenols to inhibit the formation of methemoglobin and improve the red retention rate of beef samples [1,6,28]. Compared with day 0 of storage, the *b** values of the beef samples in the T1 and T2 groups decreased significantly compared to the 12th day of storage, while the *b** values of the other three groups (T3, T4, and T5) increased significantly, indicating that the added polyphenols inhibited the enzymatic browning reaction of beef samples, and these results were in good agreement with the study by Alirezalu [1].

As an important factor, color influences consumer purchase intent because it can directly reflect the freshness of beef [25]. The surface color of beef samples from the five different treatment groups changed in various ways as the storage time was increased (Figure 1). The surface color of T1 (beef samples treated only with distilled water) showed dark spots on the sixth day of storage and completely turned black on the 12th day. In comparison to T1, the surface color of T2 (beef samples treated with GEL-CHI-GLY coating) showed dark spots on the 12th day of storage, indicating that GEL-CHI-GLY coating can shield the contact of oxygen between the environment and beef samples, effectively relieving myoglobin oxidation. These findings were consistent with those of Cardoso [11]. In addition to the barrier effect of GEL-CHI-GLY coating, the antibacterial properties of CHI are also important in preserving the color of beef samples [29]. The addition of polyphenols improved the GEL-CHI-GLY coating’s ability to protect beef color. After 12 days of storage, the surface color of T3 (beef samples treated with GEL-CHI-GLY-CA coating), T4 (beef samples treated with GEL-CHI-GLY-GA coating), and T5 (beef samples treated with GEL-CHI-GLY-RES coating) remained acceptable. Polyphenols improve the antibacterial ability of the GEL-CHI-GLY coating [24,30]: however, the antioxidant capacity of polyphenols inhibits the oxidation of beef protein [1,14].

### 3.2. pH Values and Microbiological Analysis

During storage, the pH rise in meat products is frequently caused by alkaline compounds (biogenic amines and ammonia) produced as a result of microbial activity [17,29,31]. The pH value of fresh meat is generally found to be less than 6 [1]. According to Figure 2A, the pH value of beef samples from the five different treatment groups was maintained between 5.5 and 6.0 during the initial stage of storage (day 0–day 4), but the pH value of T1 samples increased significantly to 6.27 on the sixth day of storage and reached 7.19 on the 12th day. Although the GEL-CHI-GLY coating helped to stabilize the pH of the beef samples, the pH of group T2 also reached a higher value of 6.88 after 12 days. The addition of polyphenols (CA, GA, and RES) limited the pH increase of beef samples during storage. The pH values of T3 and T4 groups remained lower and were observed to be 6.15 and 6.13 on the 12th day of storage, respectively. This suggests that CA and GA may inhibit the microbial activities of some alkali-producing compounds during beef preservation, or may help improve the antibacterial properties of GEL-CHI-GLY coating [14,24,30,32]. The decrease in pH during fresh meat preservation is primarily caused by the accumulation of carbon dioxide produced by bacterial activity (potential growth of lactic acid bacteria) [30]. Except for the control group T1, the pH value of the beef samples in the other four groups decreased during the storage process, and the pH values of the T3, T4, and T5 groups were lower than that of T2 samples (Figure 2A). This demonstrated that polyphenols (CA, GA, and RES) not only act as an active antibacterial substance to inhibit the activity of microorganisms, but they also reduce the permeability of carbon dioxide in the GEL-CHI-GLY coating film, and effectively maintain the pH stability during the storage of beef [33,34].

One of the important indicators of meat quality is the total number of microorganisms in fresh meat [4], as illustrated in Figure 2B. The TVC in beef samples from the five treatment groups gradually increased as the preservation time was extended. During the storage period, the TVC of the beef samples in group T1 increased the most, rising from 4.4 log cfu/g on day 0 to 7.6 log cfu/g on the 12th day. TVC of the T2 group also increased significantly and reached 6.6 log cfu/g on the 12th day of storage. TVC levels for T3, T4, and T5 samples remained around 6 log cfu/g after 12 days of storage, and the microorganisms in the fresh meat were still at an acceptable and safe level [6,16,25]. This showed that adding CA, GA, and RES to the GEL-CHI-GLY coating inhibited the growth of microorganisms in beef and had a similar antibacterial effect. However, CA, GA, and RES may have different inhibitory effects on different types of microorganisms. It has been reported [15] that adding GA to the CHI film had the most significant inhibitory effect on Bacillus subtilis, while [14] reported that CA had a better inhibitory effect on Escherichia coli and Pseudomonas aeruginosa in GEL film. Similarly, according to Makwana [35], RES can effectively reduce the amount of E. coli and is a promising natural antibacterial agent. In addition, some other natural substances with antibacterial activity can be used in beef preservation. For instance, the essential oil of Melaleuca alternifolia (Maiden & Betche) Cheel can inhibit the growth of Listeria monocytogenes in ground beef [36]. The presence of *Rosmarinus officinalis* L. essential oil reduced the growth of psychrophilic bacteria in beef [6].

### 3.3. Total Phenolic Content

Polyphenols are functional ingredients with antibacterial and antioxidant properties that can be used to effectively preserve a variety of aquatic products and meat [18,35,37]. As illustrated in Figure 3. The TPC in three beef samples (T3, T4, and T5) with additional polyphenols was found to be significantly higher than in the two groups without polyphenols (T1 and T2), with the TPC found to be decreased in all treated groups at varying degrees with the prolonged storage time. The TPC in group T3 samples decreased significantly, from 38.1 mg GAE/100g on day 0 to 23.1 mg GAE/100 g on the 12th day, while group T4 samples only decreased from 38.8 mg GAE/100 g on day 0 to 33.9 mg GAE/100 g on the 12th day, which was consistent with the previously reported research by Alirezalu [1], who found that oxidation caused a decrease in TPC during storage [21,34]. The different antioxidant capacities of the added polyphenols, their binding capacities, and the release degrees of different polyphenols in the GEL-CHI-GLY coating were the reasons why the TPC of varying treatment groups decreased to varying degrees [14,31,32].

### 3.4. Total Volatile Base Nitrogen

The TVB-N value has an important reference significance for determining the freshness and quality of fresh meat [23]. According to Chinese standards (GB 2707-2016) [38], the TVB-N value of fresh beef is limited to less than 150 mg/kg (Below the gray dashed line in the Figure 4). Some studies have classified beef with a TVB-N value of 150 mg/kg–200 mg/kg (Between gray and red dashed line) as secondary meat (second-grade freshness standard) [29]. As displayed in Figure 4, the TVB-N values of the beef samples in each treatment group increased as the storage time increased. The beef samples of group T1 reached 163.5 mg/kg and 262.3 mg/kg on the sixth and 9th days of storage, respectively, and no longer met the fresh beef standards (regardless of whether it is the value specified by the Chinese standard or the second-grade freshness standard) [29]. On the 9th and 12th days of storage, TVB-N values of the beef samples in group T2 reached 160.6 mg/kg and 208.7 mg/kg, respectively, exceeding the standard value. When compared to the preservation effect of a single CHI coating [29] or a GEL-CHI-GLY coating [11,24], the addition of polyphenols effectively prolonged the shelf life of beef. TVB-N values of the three groups of beef samples (T3, T4, and T5) added with polyphenols were 136.9 mg/kg, 156.5 mg/kg, and 150.9 mg/kg on the 12th day of storage, all of which met the secondary freshness standard, and the preservation effect of the T3 treatment (added with CA) was better. This indicated that CA was more effective than GA and RES at inhibiting microorganisms and enzymes from breaking down proteins and producing nitrogen-containing compounds [1,31].

### 3.5. Thiobarbituric Acid Reactive Substances

Lipid oxidation is another important factor contributing to the deterioration of fresh meat quality during the storage process. The TBARS value is frequently used to determine the degree of lipid oxidation in fresh meat [39,40]. According to Zhang [26], the TBARS values of beef had no significant relationship with sensory results, and in fact, they found that two different methods of determining TBARS values gave two different distributions, highlighting that when comparing studies, you need to ensure the methodologies are the same. Campo [41] has shown that the higher the TBARS value, the stronger the rancidity of the beef, and the worse the flavor. They considered 2.0 mg Malondialdehyde (MDA)/kg to be the acceptable threshold limit for oxidized beef. However, according to this study and the study by Zhang [25], beef samples with TBARS values less than 1.0 mg MDA/kg can maintain a lower TVB-N value, a lower pH value, and a good appearance. Limiting the TBARS value of beef to below 1.0 mg MDA/kg appears to be a more reasonable standard [42,43], although some studies considered 1.0 to be too low for beef rejection [41]. TBARS values of beef samples from the five treatments increased gradually during the storage period, as shown in Figure 5. TBARS values of four groups (T2, T3, T4, and T5) ranged from 0.71 to 0.81 mg MDA/kg on the 9th day of storage, which was significantly lower than the TBARS value of the control group T1 (1.1 mg MDA/kg), indicating that beef samples of control group T1 were approaching rancidity on the 9th day, while the remaining four groups (T2, T3, T4, and T5) were still acceptable. On the 12th day of storage, TBARS values of the two groups of beef samples, to which either CA or RES were added, remained low at 0.76 mg MDA/kg and 0.78 mg MDA/kg, respectively, indicating that the addition of CA and RES increased the anti-lipid oxidation ability of the GEL-CHI-GLY coating [44]. There are typically two reasons why the coating film could inhibit beef lipid oxidation. The first is the oxygen barrier effect of the formed coating [12], and the second is the effect of inhibiting the growth of microorganisms and the role that anti-oxidation plays by adding active ingredients [7,29,45]. Fu [14] has shown that GEL films containing CA had better antioxidant effects than neat GEL films. Figure 5 also illustrates that CA, GA, and RES were effective additives that can be used in GEL-CHI-GLY coating to preserve beef preservation.

### 3.6. Texture Profile Analysis

The texture is an important indicator as it intuitively reflects the tenderness of the meat. The effect of the different treatments on the texture of the beef was determined by measuring the changes in the hardness, springiness, and chewiness of beef samples during the storage process, as shown in Table 2. The hardness, springiness, and chewiness of beef samples from all treatment groups decreased with the increase in storage time, as expected, and in agreement with the report by Zhang [25]. The hardness of beef reflects the tenderness and has been linked to its fat content and muscle myofibrils [3,46,47]. The degree of decrease in hardness of T3 beef samples from day 0 to the 12th day of storage was significantly less than in the other four groups (T1, T2, T4, and T5), indicating that CA may have a protective effect on the beef muscle myofibrils and prevent degradation. The research results of Cheng [48] and Zhao [49] both showed that polyphenols can stabilize the protein in the muscle. The research of Sun [50] also revealed that CA is the main component of young apple polyphenols, and has an antiseptic effect, which can delay the oxidative degradation of myofibrillar proteins, and thus maintain the texture of surimi. On the 12th day of storage, the reduction in chewiness of T3 beef samples was also the smallest, although its value (115.7 g) was the smallest among the four GEL-CHI-GLY treated groups. This showed that CA may also play a better role in maintaining the texture of beef. Springiness is the ability of a sample to return to its original shape after removing the external force [25]. As shown in Table 2, the springiness of T4 beef samples was 0.7 on the 12th day of storage, which was significantly higher than the other four groups, indicating that the GA maintained better elasticity of beef. The hardness, springiness, and chewiness of the four groups (T2, T3, T4, and T5) of beef samples treated with GEL-CHI-GLY coating were significantly higher than those of the control group T1 on the 12th day of storage. This demonstrated that the GEL-CHI-GLY coating treatment preserves beef texture.

### 3.7. Sensory Evaluation

Table 3 shows the sensory (color, odor, and texture) evaluation results of beef samples from the five treatment groups during the storage period. On day 0 of storage, there were no significant differences (*p* > 0.05) between the samples’ color, odor, and texture scores from all treatment groups. This indicated that the GEL-CHI-GLY coating and the addition of polyphenols had no negative effect on the sensory properties of beef meat, as was the case with ε-polylysine and stinging nettle extract-treated beef [1]. After 12 days of storage, the sensory evaluation index scores of beef samples from all treatment groups decreased significantly (*p* < 0.05), and all T1 and T2 sensory evaluation index scores were less than 5.4 (acceptability limit of the sensory evaluation used in this study). The three polyphenol addition treatment groups (T3, T4, and T5) significantly reduced the color deterioration, the occurrence of off-odors, and texture (surface stickiness) of beef during storage, indicating that the addition of polyphenols enhances the antibacterial and antioxidant activities of the GEL-CHI-GLY coating, preserving the color, odor, and texture of beef within acceptable limits after 12 days of storage.

## 4. Conclusions

In this study, chlorogenic acid (CA), gallic acid (GA), and resveratrol (RES) were added to a gelatin (GEL)-chitosan (CHI)-glycerol (GLY) edible coating, and their effect on fresh beef preservation was investigated. The results showed that adding polyphenols to the GEL-CHI-GLY edible coating improved its ability to preserve beef and prolonged the shelf life by 3–6 days, compared with uncoated beef and GEL-CHI-GLY coated beef. The CA, in particular, displayed the best improvement effect on the GEL-CHI-GLY edible coating on beef preservation, which better protected the color of beef and delayed the increase of beef TVB-N value, despite being released faster (TPC drops the most) during beef preservation. On the 12th day of storage, the GEL-CHI-GLY-GA treated beef had the lowest TBARS values and total bacterial counts. The color, odor, and texture of beef from all polyphenol-added treatment groups remained acceptable after 12 days of storage. This showed that CA, GA, and RES could be used as effective additives in GEL-CHI-GLY edible coatings to improve the preservation ability of beef. This study preliminarily determined that the edible coating of GEL-CHI-GLY complexed with CA, GA, and RES has a beneficial effect on fresh beef preservation. In the next study, the dominant spoilage bacteria in beef can be isolated, and the inhibitory effect of different polyphenols on the dominant spoilage bacteria can be explored to further improve the research. This study has also suggested the application of more natural antioxidant-active substances in meat preservation.

## Figures and Tables

**Figure 1 foods-11-03813-f001:**
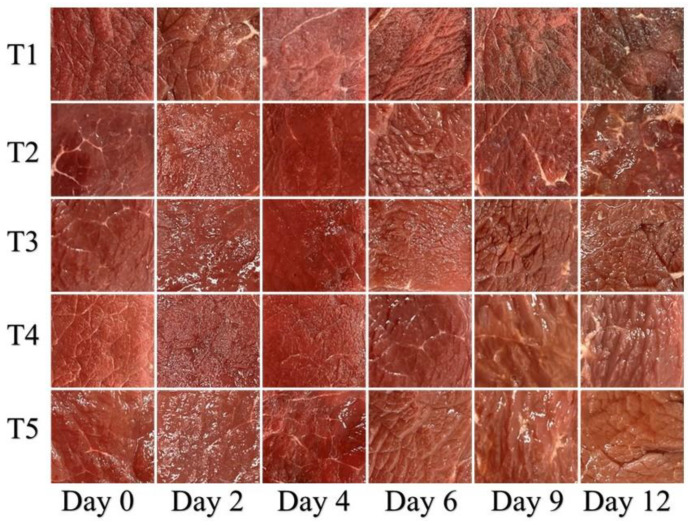
Images of beef samples from different treatment groups during storage. T1: Control, beef samples immersed in distilled water; T2: Beef samples treated with GEL-CHI-GLY edible coating; T3: Beef samples treated with GEL-CHI-GLY-CA edible coating; T4: Beef samples treated with GEL-CHI-GLY-GA edible coating; and T5: Beef samples treated with GEL-CHI-GLY-RES edible coating.

**Figure 2 foods-11-03813-f002:**
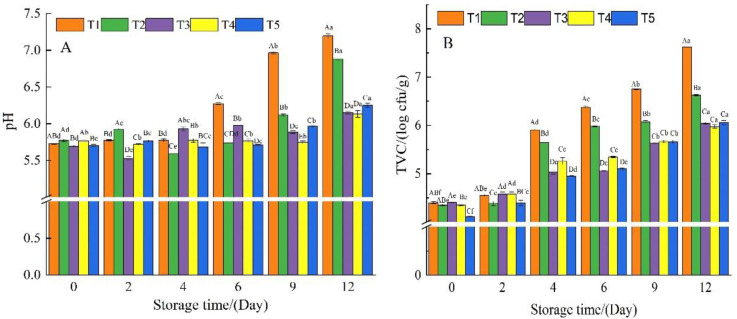
The pH (**A**) and total viable count (**B**) of beef samples from different treatment groups during storage. T1: Control, beef samples immersed in distilled water; T2: Beef samples treated with GEL-CHI-GLY edible coating; T3: Beef samples treated with GEL-CHI-GLY-CA edible coating; T4: Beef samples treated with GEL-CHI-GLY-GA edible coating; and T5: Beef samples treated with GEL-CHI-GLY-RES edible coating. ^a–e^ Values in the same row bearing different superscripts are different (*p* < 0.05). ^A–E^ Values in the same column bearing different superscripts are different (*p* < 0.05).

**Figure 3 foods-11-03813-f003:**
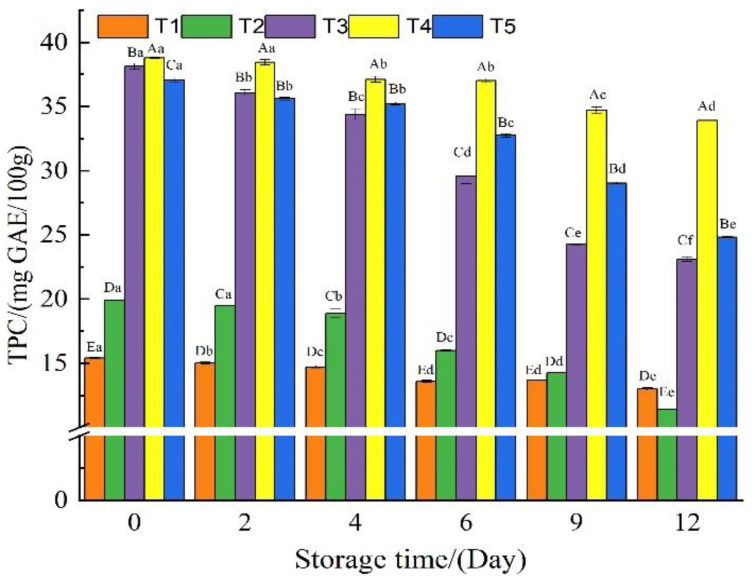
The total phenolic content of beef samples from different treatment groups during storage. T1: Control, beef samples immersed in distilled water; T2: Beef samples treated with GEL-CHI-GLY edible coating; T3: Beef samples treated with GEL-CHI-GLY-CA edible coating; T4: Beef samples treated with GEL-CHI-GLY-GA edible coating; and T5: Beef samples treated with GEL-CHI-GLY-RES edible coating. ^a–e^ Values in the same row bearing different superscripts are different (*p* < 0.05). ^A–E^ Values in the same column bearing different superscripts are different (*p* < 0.05).

**Figure 4 foods-11-03813-f004:**
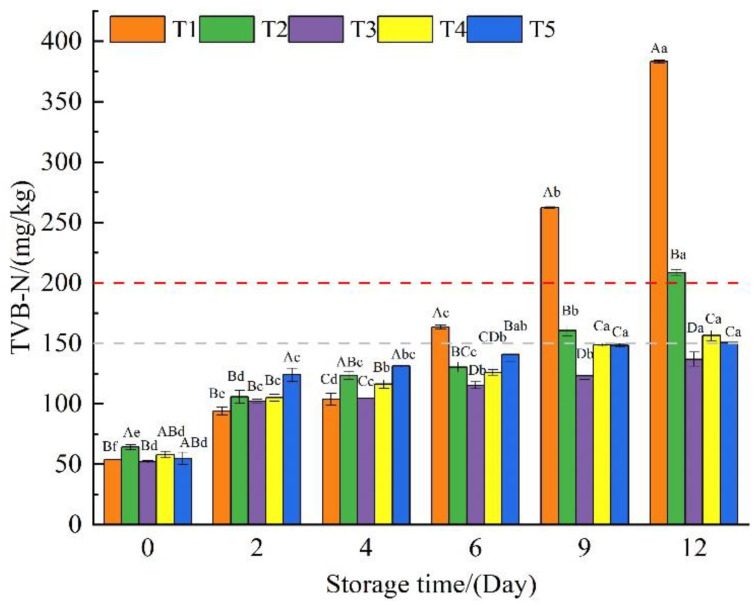
The total volatile base nitrogen of beef samples from different treatment groups during storage. T1: Control, beef samples immersed in distilled water; T2: Beef samples treated with GEL-CHI-GLY edible coating; T3: Beef samples treated with GEL-CHI-GLY-CA edible coating; T4: Beef samples treated with GEL-CHI-GLY-GA edible coating; and T5: Beef samples treated with GEL-CHI-GLY-RES edible coating. ^a–f^ Values in the same row bearing different superscripts are different (*p* < 0.05). ^A–D^ Values in the same column bearing different superscripts are different (*p* < 0.05).

**Figure 5 foods-11-03813-f005:**
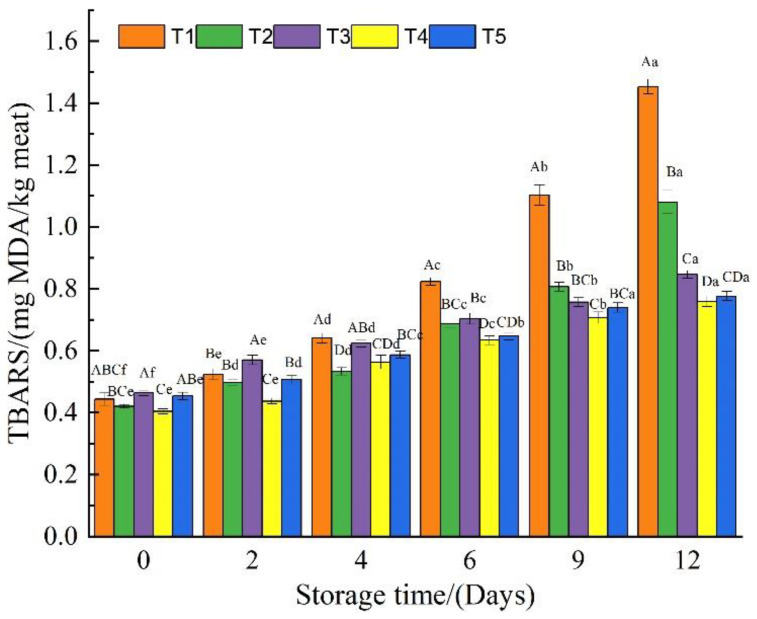
The thiobarbituric acid reactive substances of beef samples from different treatment groups during storage. T1: Control, beef samples immersed in distilled water; T2: Beef samples treated with GEL-CHI-GLY edible coating; T3: Beef samples treated with GEL-CHI-GLY-CA edible coating; T4: Beef samples treated with GEL-CHI-GLY-GA edible coating; and T5: Beef samples treated with GEL-CHI-GLY-RES edible coating. ^a–f^ Values in the same row bearing different superscripts are different (*p* < 0.05). ^A–D^ Values in the same column bearing different superscripts are different (*p* < 0.05).

**Table 1 foods-11-03813-t001:** The color of beef samples from different treatment groups during storage (mean ± S.E.).

	Beef	Storage Time (Day)					
		0	2	4	6	9	12
*L**	T1	41.7 ± 0.5 ^Ba^	40.6 ± 0.5 ^Bab^	41.8 ± 1.0 ^ABa^	39.0 ± 0.5 ^Bbc^	37.0 ± 0.6 ^Bc^	37.6 ± 0.5 ^CDc^
	T2	39.7 ± 0.5 ^Cbc^	43.5 ± 0.4 ^Aa^	40.2 ± 0.8 ^Bb^	42.0 ± 0.4 ^Aa^	38.0 ± 0.3 ^Bc^	36.6 ± 0.4 ^Dd^
	T3	43.3 ± 0.8 ^Aba^	38.1 ± 0.3 ^Cc^	39.9 ± 0.2 ^Bbc^	41.7 ± 0.6 ^Aab^	42.5 ± 0.8 ^Aa^	42.1 ± 0.7 ^ABa^
	T4	44.6 ± 0.4 ^Aa^	38.6 ± 0.0 ^Cc^	40.6 ± 0.1 ^ABbc^	41.5 ± 0.2 ^Ab^	43.9 ± 0.4 ^Aa^	39.5 ± 1.6 ^BCbc^
	T5	44.3 ± 0.7 ^Aa^	38.6 ± 0.1 ^Cb^	42.6 ± 0.5 ^Aa^	43.1 ± 0.7 ^Aa^	43.4 ± 0.5 ^Aa^	42.7 ± 0.4 ^Aa^
*a**	T1	14.8 ± 0.2 ^Aa^	13.3 ± 0.0 ^Ab^	12.0 ± 0.3 ^Cc^	11.4 ± 0.2 ^Cc^	9.2 ± 0.3 ^Cd^	7.4 ± 0.1 ^Ce^
	T2	12.9 ± 0.1 ^Bb^	12.7 ± 0.2 ^Bb^	14.2 ± 0.3 ^Aa^	13.2 ± 0.2 ^ABb^	11.1 ± 0.1 ^Bc^	9.5 ± 0.3 ^Bd^
	T3	12.8 ± 0.2 ^Bb^	11.8 ± 0.1 ^Ccd^	13.8 ± 0.2 ^Ba^	12.8 ± 0.1 ^Bb^	12.3 ± 0.2 ^Abc^	10.5 ± 0.2 ^Ae^
	T4	12.6 ± 0.1 ^Bb^	12.5 ± 0.1 ^Bb^	12.4 ± 0.1 ^Cb^	13.5 ± 0.1 ^Aa^	12.8 ± 0.4 ^Ab^	10.1 ± 0.1 ^ABc^
	T5	12.8 ± 0.3 ^Bb^	12.4 ± 0.2 ^Bb^	14.1 ± 0.2 ^Aa^	11.4 ± 0.0 ^Cc^	10.5 ± 0.1 ^Bd^	9.8 ± 0.4 ^ABe^
*b**	T1	7.2 ± 0.2 ^Aa^	6.5 ± 0.0 ^Ab^	6.2 ± 0.2 ^Ab^	6.1 ± 0.1 ^Cb^	4.4 ± 0.1 ^Cc^	3.7 ± 0.1 ^Dd^
	T2	6.2 ± 0.3 ^Bbc^	5.7 ± 0.1 ^Bcd^	6.7 ± 0.2 ^Aab^	6.8 ± 0.1 ^ABa^	6.1 ± 0.1 ^Bc^	5.3 ± 0.2 ^Cd^
	T3	5.6 ± 0.2 ^Ccd^	5.4 ± 0.1 ^Bd^	6.5 ± 0.1 ^Ab^	7.2 ± 0.2 ^Aa^	6.1 ± 0.1 ^Bbc^	7.7 ± 0.3 ^Aa^
	T4	5.3 ± 0.0 ^Cd^	5.7 ± 0.0 ^Bcd^	6.4 ± 0.1 ^Aab^	6.0 ± 0.0 ^Cbc^	6.2 ± 0.3 ^Bab^	6.7 ± 0.2 ^Ba^
	T5	5.2 ± 0.2 ^Cd^	5.6 ± 0.2 ^Bcd^	5.5 ± 0.2 ^Bd^	6.3 ± 0.1 ^BCbc^	7.2 ± 0.4 ^Aa^	6.4 ± 0.2 ^Bb^

T1: Control, beef samples immersed in distilled water; T2: Beef samples treated with GEL-CHI-GLY edible coating; T3: Beef samples treated with GEL-CHI-GLY-CA edible coating; T4: Beef samples treated with GEL-CHI-GLY-GA edible coating; and T5: Beef samples treated with GEL-CHI-GLY-RES edible coating; *L**: Lightness; *a**: Redness; *b**: Yellowness. ^a–e^ Values in the same row bearing different superscripts are different (*p* < 0.05). ^A–D^ Values in the same column bearing different superscripts are different (*p* < 0.05).

**Table 2 foods-11-03813-t002:** The TPA texture of beef samples from different treatment groups during storage (mean ± S.E.).

	Beef	Storage Time (Day)					
		0	2	4	6	9	12
Hardness/(g)	T1	871.4 ± 27.7 ^Ca^	731.5 ± 38.4 ^Db^	537.2 ± 35.0 ^Dc^	530.0 ± 5.1 ^Cc^	316.8 ± 34.3 ^Dd^	282.0 ± 10.9 ^Ee^
	T2	954.2 ± 44.3 ^Aa^	784.1 ± 43.6 ^Cb^	573.6 ± 28.2 ^Cc^	501.8 ± 56.5 ^CDd^	416.1 ± 40.8 ^Ce^	370.5 ± 10.0 ^Cf^
	T3	699.9 ± 23.8 ^Da^	512.9 ± 16.9 ^Eb^	481.3 ± 62.3 ^Ec^	438.7 ± 21.4 ^Dd^	409.0 ± 19.0 ^Ce^	349.2 ± 8.0 ^Df^
	T4	949.0 ± 44.1 ^Aa^	836.8 ± 71.5 ^Bb^	749.2 ± 26.7 ^Ac^	668.6 ± 117.2 ^Ad^	488.9 ± 60.3 ^Be^	418.7 ± 34.3 ^Bf^
	T5	912.6 ± 89.2 ^Ba^	899.5 ± 31.1 ^Aa^	624.0 ± 19.3 ^Bb^	586.6 ± 21.6 ^Bc^	585.5 ± 43.1 ^Ac^	454.1 ± 70.6 ^Ad^
Springiness	T1	0.78 ± 0.08 ^Ba^	0.65 ± 0.07 ^Db^	0.61 ± 0.04 ^Dc^	0.56 ± 0.02 ^Ed^	0.53 ± 0.03 ^Dde^	0.47 ± 0.01 ^De^
	T2	0.85 ± 0.08 ^Aa^	0.80 ± 0.09 ^Bab^	0.75 ± 0.05 ^ABb^	0.62 ± 0.04 ^Dc^	0.58 ± 0.11 ^Ccd^	0.55 ± 0.04 ^Cd^
	T3	0.74 ± 0.05 ^Ca^	0.70 ± 0.02 ^Cab^	0.67 ± 0.01 ^Cb^	0.65 ± 0.07 ^Cbc^	0.63 ± 0.05 ^Bcd^	0.58 ± 0.05 ^Bd^
	T4	0.85 ± 0.09 ^Aa^	0.84 ± 0.06 ^Aa^	0.73 ± 0.02 ^Bb^	0.72 ± 0.05 ^Bb^	0.72 ± 0.05 ^Ab^	0.70 ± 0.05 ^Ab^
	T5	0.86 ± 0.04 ^Aa^	0.85 ± 0.05 ^Aa^	0.79 ± 0.04 ^Ab^	0.78 ± 0.08 ^Ab^	0.71 ± 0.03 ^Ac^	0.58 ± 0.07 ^Bd^
Chewiness/(g)	T1	232.2 ± 17.0 ^Ba^	227.8 ± 5.3 ^Bab^	171.2 ± 32.0 ^Db^	167.5 ± 21.3 ^CDbc^	107.9 ± 21.3 ^Ec^	65.9 ± 9.6 ^Cd^
	T2	364.9 ± 29.6 ^Aa^	215.1 ± 28.6 ^BCb^	209.8 ± 32.8 ^Cbc^	199.7 ± 29.4 ^Ccd^	197.1 ± 28.9 ^Ccd^	116.6 ± 27.9 ^Bd^
	T3	233.3 ± 43.6 ^Ba^	198.0 ± 8.4 ^Cb^	194.5 ± 4.5 ^CDb^	154.5 ± 17.6 ^Dc^	138.7 ± 25.1 ^Dd^	115.7 ± 8.0 ^Be^
	T4	378.2 ± 59.0 ^Aa^	338.7 ± 20.8 ^ABb^	310.5 ± 26.9 ^Ac^	294.7 ± 31.1 ^Ad^	213.6 ± 40.1 ^Be^	178.0 ± 3.9 ^Af^
	T5	375.1 ± 56.6 ^Aa^	345.4 ± 35.6 ^Ab^	260.2 ± 31.7 ^Bc^	256.7 ± 50.9 ^Bc^	243.7 ± 8.0 ^Ad^	183.3 ± 21.2 ^Ae^

T1: Control, beef samples immersed in distilled water; T2: Beef samples treated with GEL-CHI-GLY edible coating; T3: Beef samples treated with GEL-CHI-GLY-CA edible coating; T4: Beef samples treated with GEL-CHI-GLY-GA edible coating; T5: Beef samples treated with GEL-CHI-GLY-RES edible coating. ^a–f^ Values in the same row bearing different superscripts are different (*p* < 0.05). ^A–E^ Values in the same column bearing different superscripts are different (*p* < 0.05).

**Table 3 foods-11-03813-t003:** The sensory evaluation of beef samples from different treatment groups during storage. (mean ± S.E.).

Sensory	Color		Odor		Texture	
	0	12	0	12	0	12
T1	8.3 ± 0.1 ^Aa^	2.2 ± 0.2 ^Cb^	8.3 ± 0.2 ^Aa^	1.9 ± 0.2 ^Cb^	8.3 ± 0.1 ^Aa^	1.5 ± 0.1 ^Cb^
T2	8.0 ± 0.1 ^Aa^	4.4 ± 0.2 ^Bb^	8.5 ± 0.1 ^Aa^	3.5 ± 0.2 ^Bb^	7.9 ± 0.1 ^Aa^	3.7 ± 0.1 ^Bb^
T3	7.8 ± 0.1 ^Aa^	6.2 ± 0.2 ^Ab^	8.4 ± 0.1 ^Aa^	6.2 ± 0.1 ^Ab^	8.03 ± 0.1 ^Aa^	5.7 ± 0.2 ^Ab^
T4	8.0 ± 0.2 ^Aa^	6.2 ± 0.1 ^Ab^	8.4 ± 0.0 ^Aa^	6.4 ± 0.1 ^Ab^	8.2 ± 0.2 ^Aa^	5.8 ± 0.2 ^Ab^
T5	8.3 ± 0.2 ^Aa^	6.3 ± 0.1 ^Ab^	8.4 ± 0.2 ^Aa^	6.3 ± 0.1 ^Ab^	8.3 ± 0.1 ^Aa^	6.1 ± 0.1 ^Ab^

T1: Control, beef samples immersed in distilled water; T2: Beef samples treated with GEL-CHI-GLY edible coating; T3: Beef samples treated with GEL-CHI-GLY-CA edible coating; T4: Beef samples treated with GEL-CHI-GLY-GA edible coating; and T5: Beef samples treated with GEL-CHI-GLY-RES edible coating. ^a,b^ Different lowercase letters indicate a significant (*p* < 0.05) difference between storage days. ^A–C^ Different capital letters indicate a significant (*p* < 0.05) difference between treatments.

## Data Availability

The data presented in this study are available on request from the corresponding author.

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
