# Peer review of "Influence of Gelatin-Chitosan-Glycerol Edible Coating Incorporated with Chlorogenic Acid, Gallic Acid, and Resveratrol on the Preservation of Fresh Beef"

_foods, 2022, doi:10.3390/foods11233813_

Round 1
Reviewer 1 Report
Dear Editor,
Research on this topic is current. An adequate amount of information has been presented in the introduction, and the researchers have clearly outlined their objectives. Materials and methods have been presented in an understandable manner, following a specified sequence. Visual enrichment has been used to present research results. There are a few references that need to be provided in the discussion, but the discussion is satisfactory otherwise. There is a clear and understandable conclusion. It is therefore my recommendation that minor changes are made.
Minor comments:
L5-11: It should be rewritten in accordance with journal writing guidelines
It must be achieved to collaborate while giving “P” values in the text. Some are italicized and some are straight.
“CFU” should be corrected as “cfu” in the whole text.
Minor comments for “Introduction” section
L50-51: “Rumex tingitanus”, “Syzygium aromaticum”, “Cinnamomum cassia” , “Origanum vulgare” , “Brassica nigra” should be corrected as “Rumex tingitanus L.”, “Syzygium aromaticum (L.) Merrill et L.M. Perry”, “Cinnamomum cassia (L.) J.Presl”, “ Origanum vulgare L.”,” Brassica nigra L.” (An author's name should also be included in binomial nomenclature).
L55-56: “Chlorogenic acid (CA), gallic acid (GA), and resveratrol (RES)…” instead of “Chlorogenic acid, gallic acid, and resveratrol…” (The abbreviation should be used since this is the first time it is used).
L58: “…CA…” instead of “…chlorogenic acid…”
L60: “Similarly, GA [24] and RES [16]…” instead of “Similarly, gallic acid [24] and resveratrol [16]….”
L62: “gelatin (GEL)-chitosan (CHI)-glycerol (GLY) edible coating” instead of “gelatin-chitosan-gelatin edible coating”
L64-65: “(CA, GA, and RES) in a GEL-CHI-GLY edible coating” instead of (chlorogenic acid, gallic acid, and resveratrol) in a gelatin-chitosan-glycerin edible coating
Minor comments for “Materials and Methods” section
L74: “The GA, CA, RES, CHI, GEL, GLY…” instead of “Gallic acid (GA), chlorogenic acid (CA), resveratrol (RES), chitosan, gelatin, and glycerol”.
L77: “GEL” instead of “gelatin”
L79: “CHI” instead of “chitosan”
L79: “GLY” instead of “glycerol”
L81-82: “ GA, CA, and RES” instead of “gallic acid, chlorogenic acid, and resveratrol”
L83: “GEL, CHI, GLY” instead of “gelatin, chitosan, glycerol”
L94: “….pH, total viable count (TVC), total phenolic content (TPC), total volatile base nitrogen (TVB-N)…” instead of “….pH, total viable count, total phenolic content, total volatile base nitrogen…”
L101: “L*”, “a*, ”b*” should be italicized
L115: “cfu” instead of “CFU”
L126, L132, L141, L266, L286, L310, L344: In these sub-section titles, abbreviations should not be given. It would be more appropriate to provide the abbreviation in the first sentence following the title.
Minor comments for “Results and Discussion” section
L169: “L*”, “a*, ”b*” should be italicized
L191: It would be better if the "Lightness, L*; redness, a*; yellowness, b*" terms are given as footnotes at the bottom of the table.
L193: There should be a clear description of the treatment that the trial groups received; abbreviations should be avoided.
L201: “6th” instead of “sixth”
L202: “GEL-CHI-GLY” instead of “gelatin-chitosan-glycerol”
L206, L208, L213, L224, L231, L238, L278, L298, L332, L361, L367, L380: “GEL-CHI” instead of “gelatin-chitosan”
L207: “CHI” instead of “chitosan”
L209: “GEL-CHI-GLY-CA” instead of “gelatin- chitosan-glycerol-chlorogenic acid”
L210: “GEL-CHI-GLY-GA” instead of “gelatin- chitosan-glycerol-gallic acid”
L211: “GEL-CHI-GLY-RES” instead of “gelatin- chitosan-glycerol- resveratrol”
L216: Under the figure, the contents of the experimental groups should be described.
L226, L236: “CA, GA, and RES” instead of “chlorogenic acid, gallic acid, and resveratrol”
L229: “CA and GA” instead of “chlorogenic acid and gallic acid”
L236: “group” instead of “samples”
L242: “The TVC” instead of “The total viable count (TVC)” (previously abbreviated)
L249: “CA, GA, RES to the GEL-CHI” instead of “chlorogenic acid, gallic acid, and resveratrol to the gelatin-chitosan”
L:251: “CA, GA, RES” instead of “chlorogenic acid, gallic acid, and resveratrol”
L253-255: As they are used for the first time, B.subtilis, E.coli, and P.aeruginosa should be written clearly. The following terms should be abbreviated: gallic acid, chlorogenic acid, chitosan, and gelatin.
L255: “RES” instead of “resveratrol”
L255: “Melaleuca alternifolia” instead of “Melaleuca alternifolia (Maiden & Betche) Cheel”
L260: “CFU” should be corrected as “cfu” in Figure2B.
L264: There should be a clear description of the treatment that the trial groups received; abbreviations should be avoided.
L269-272: “TPC” instead of “total phenolic content”
L288: It is recommended that the Chinese standard be added to the reference list.
L295: The previous references should be referred to one more time.
L298:”CHI” instead of “chitosan”
L303-305:”CA”, “GA”, and “RES” instead of ”chlorogenic acid”, “gallic acid” and “resveratrol”.
L308: There should be a clear description of the treatment that the trial groups received; abbreviations should be avoided. Below the figure, it should be noted according to which standard the dashed lines in gray and red are referenced.
L318: “2.0 mg Malondialdehyde(MDA)/kg instead of “2.0 mg MDA/kg”
L330-332:”CA” and “RES” instead of ”chlorogenic acid” and “resveratrol”
L336, L337:”GEL” instead of ”gelatin”
L337, L353, L361, L402:”CA” instead of ” chlorogenic acid”
L338, L409: “CA, GA, RES” instead of “chlorogenic acid, gallic acid, and resveratrol”
L338, L369, L389, L398, L400, L402, L403, L409: “GEL-CHI-GLY” instead of “gelatin-chitosan-glycerin”
L342: There should be a clear description of the treatment that the trial groups received; abbreviations should be avoided. An explanation of what TBARS and MDA are should be provided.
L350: There must be more than one reference
L365:”GA” instead of ”gallic acid”
L370: “texture” instead of “Texture”. The table should be given at the beginning of the next page so that it can be viewed in its entirety. There should be a clear description of the treatment that the trial groups received; abbreviations should be avoided. Furthermore, the distance between the columns should be considered. As a result of the journal's format, the table may expand to the left.
L393: There should be a clear description of the treatment that the trial groups received; abbreviations should be avoided

Author Response
Dear reviewer
We greatly appreciate your active consideration and helpful suggestions for revision of our manuscript. We have made revisions to the manuscript based on your kind suggestion, the corresponding responses to your comments are shown in red, the page, row and figure or table numbers refer to the revised manuscript, and the changes we have made are shown throughout the revised manuscript with Yellow highlighted. If you have any other questions, please let me know as soon as possible.
Best Regards,
Zou Jinhao

Reviewer 2 Report
The article entitled “influence of gelatin-chitosan-glycerol edible coating incorporated with chlorogenic acid, gallic acid, and resveratrol on the preservation of fresh beef”, has been written well. The article can be modified in few major revisions
In the section treatment, beef was divide into 90 pieces. It is better to write how the author keep the products to be safet during the treatment, since this steps is the critical steps, causing contamination that might affects the beef shelf life.
Line 90-91, the beef of control group was immersed in distilled water. Why is the beef immersed in the water? It increases the water activity of the beef and causes the cross-contamination, and then reduces the shelf life of the beef. The beef should not be immersed in the water and directly placed in single use plastic lunch boxes.
Besides, the antimicrobial activity of the added antimicrobials affects the beef, the characteristic of the coating surface, water absorption, oxygen and water barrier of coating might also influence the beef shelf life. In this article, Author mentioned about this effects, such as lines 333-335. How did Author consider this effects of coating characteristics? Is there any possibility that the phenolic compounds affect the coating characteristics?
Is there any information about the thickness of the film? If yes, it is better to present in this work as well
The author wrote the abbreviations of gelatin, glycerol and chitosan as GEL, GLY, and CHI, respectively. However, the abbreviation only mentioned in the first time in the abstract. It should also be stated in the body of the manuscript; Introduction or method.
Instead of Bar charts, I suggest the Figures for pH, TPC, TVB-N, TBARS, presented in line chart. It can be easily to follow the increase or decrease as a function of time.
Author Response

(The authors gave the same response as above.)

Reviewer 3 Report
The manuscript is dealing with the application of an edible coating activated with three different polyphenols. The activated edible coatings were applied to the fresh beef meat and their effects on beef preservation were investigated. The study has some practical applications but it needs revision.
The sensory analysis description is not very clear. It was performed on raw or cooked meat?
What means “texture”? How it was determined?
Moreover, the number of the panelist for the sensory analysis is not enough. Explain the condition of lighting and the sensory room that was selected for sensory evaluation. How many of the panelists were men and women? what was the age range? how did you recruit them?
Did you obtain any IRB approval to conduct this evaluation?
Another fundamental parameter to investigate, related to the storage of meat and meat products, is the Water Holding Capacity. It would have been useful to evaluate it and link it to the results obtained from the TPA analysis
According to the TVC results, the bacterial population of T1 sample exceeds the 7 log CFU/g which is not suitable for human consumption. How did you serve these samples to the panelists?
The Authors are invited to check the English.
Author Response

(The authors gave the same response as above.)

Round 2
Reviewer 2 Report
The comment has been addressed well.